# Factors Associated with Practice of Chemical Pesticide Use and Acute Poisoning Experienced by Farmers in Chitwan District, Nepal

**DOI:** 10.3390/ijerph18084194

**Published:** 2021-04-15

**Authors:** Simrin Kafle, Abhinav Vaidya, Bandana Pradhan, Erik Jørs, Sharad Onta

**Affiliations:** 1Nepal Public Health Foundation, Kathmandu 44600, Nepal or vaidya_abhinav@nphfoundation.org (A.V.); or onta_sharad@nphfoundation.org (S.O.); 2Institute of Medicine, Tribhuwan University, Kathmandu 44600, Nepal; bandana@reachpuba.org; 3Clinic of Occupational Medicine, Clinical Institute, University of Southern Denmark, 5000 Odense, Denmark; erik.dialogos@gmail.com or

**Keywords:** pesticides, safety measures, acute poisoning, Nepal

## Abstract

In view of increasing irrational use and unsafe handling of pesticides in agriculture in Nepal, a descriptive cross-sectional study was conducted to assess the practice of chemical pesticide use and acute health symptoms experienced by farmers. A total of 790 farmers from the Chitwan district were randomly selected for the study. X^2^ test, T-test, and Multiple Logistic Regression were used for analysis. Among the farmers, 84% used exclusively chemical pesticide. Farmers with better knowledge on pesticide handling were 8.3 times more likely to practice safe purchasing, four times more likely to practice safe mixing and spraying, and two times more likely to practice safe storage and disposal. Similarly, perception/attitude of farmers about chemical pesticide policy and market management was significantly associated with the practice of farmers during purchasing, mixing and spraying, and storage and disposal. Among the users of chemical pesticides, 18.7% farmers experienced one or more pesticide related acute symptoms of health problems during the previous 12 months. Farmers with unsafe practices of pesticide handling were two times more likely to suffer from acute poisoning. It is concluded that knowledge about pesticide handling and favorable perception/attitude on pesticide policy and market management are the predictors of safe use of pesticide.

## 1. Introduction

Increasing pesticide use in farming has become a global public health issue, affecting middle- and low-income countries [1]. Global pesticide use increased by 46% between 1996 and 2016 [2,3]. The total world land area is 13.5 billion ha, of which 4.9 billion ha is agricultural land [4]. In 2016, the total amount of active ingredients in pesticides used in agriculture was 4.1 million tons worldwide [4].

In Nepal, the consumption average weight of active ingredients of pesticides applied is 396 g/ha [5]. This amount is lower in comparison to other countries (for example, India 0.5 kg/ha, China 14 kg/ha) [6], but due to irrational use and unsafe handling, the issue of pesticide use in agriculture farming is becoming a growing public health concern [7,8]. Moreover, its use in Nepal is concentrated in relatively few provinces and also increasing by about 20% per year [5]. Of the total pesticides imported in the country, more than 90% is used in vegetable farming [5].

The Joint Food and Agriculture Organization and World Health Organization meeting on pesticide residues has established Maximum Residual Limits (MRLs) for pesticides in foods to ensure pesticide exposure through eating food over the lifetime will not lead to adverse effects on health [9]. However, evidence suggests that many developing countries lack a pesticide residue measurement system in place to effectively monitor the permissible limits of pesticides in foods before entering into the market [3,10], thus jeopardizing public health.

Health problems associated with pesticide include poisonings due to suicide attempt, contaminated food, and unintended and occupational accidents and injuries leading to deaths [11]. Pesticide use is also linked to several acute and chronic health problems, more noticeable in developing countries including Nepal [12,13,14]. In Nepal, the issue of pesticide and its effect on human health has been stipulated in National Health Policy 2020 for the first time (policy number 6.12, strategy 6.12.5) stating that the state will control and regulate the use of pesticides in foods affecting human health [15]. However, public health programs acting on this policy are yet to be designed and implemented [16].

Farmers are the ones who are most likely to be exposed to pesticides [17,18], and despite the increasing import and use of pesticide in the country, studies about the practice of farmers on the issue and their experience of health problems while handling them are still scant [19]. In view of this, the present study was conducted with the objective to assess knowledge, attitude/perception, and practice (KAP) of farmers and their experience of poisoning symptoms after exposure to pesticides with the aim to generate evidence to reduce the harm associated with pesticide use.

## 2. Materials and Methods

### 2.1. Setting, Study Design, and Site

The descriptive cross-sectional study was conducted in Chitwan district, one of the 77 districts of Nepal, and covered all of its seven municipalities. Located at the south central part of the country in Bagmati Province, the district is well known for high production of commercial vegetables coupled with easy availability of chemical pesticides, legally or illegally imported through the porous borders [11]. The duration of the study was from October 2019 to May 2020. The climate of Chitwan is hot and humid tropical climate.

### 2.2. Study Population and Sampling

Farmers engaged in crop production were included in the study. The sampling frame for farmer selection was obtained from District Cooperative Office (DCO), Chitwan, Nepal. Farmers engaged in agriculture cooperatives registered in DCO provided the sampling frame.

For farmers, sample size was estimated using the formula and calculation as given, n = Nz^2^pq/e^2^(N − 1) + z^2^pq [20], where N represents the total number of crop growing farmers in Chitwan which was 42,548, z = percentiles of the standard normal distribution corresponding to 95% confidence level which is equal to 1.96, *p* = Percent of farmers using pesticides in their farm, was assumed 50, q = 1 − *p*, percent of farmers not using pesticides in their farm, and e denotes margin of error = 5 at 95% confidence level. Therefore, using the formula, n = 42,548(1.96)^2^ × 50 × 50/(5)^2^(42,548 − 1) + (1.96)^2^ × 50 × 50 = 380.73 and adding design effect = 380.73 × 2 = 761.46, and assuming non-response rate as 4%, the total sample size estimated for the study was 791.92~792.

For the sample selection, each municipality was considered as a cluster. There is one metropolitan city, five urban municipalities, and one rural municipality in Chitwan district. Farmers’ population in different municipalities was first identified, and then we applied probability proportional to size sampling to calculate sample size for each cluster from the total 792. Having listed the names of all farmers in an excel sheet, we used systematic random sampling.

### 2.3. Data Collection Tools

Data collection tools were developed reviewing relevant literature from the subject area being based on indicators considered through literature [18,21,22] to assess the practice of farmers about safe handling and associated factors. All the questions were close ended, developed in Nepali, and translated into English and then back translated into Nepali in order to check for its reliability. Interviewers were provided three days training on objectives, methods, and process of data collection, and it was accomplished under the supervision of the principal investigator. The tools were pre-tested in 10% of total sample size in the adjoining district and any changes required were made. Face to face interviews were conducted to collect data from farmers. In addition, observation was done to their storehouse/place to verify their practice during storage. Additionally, the verbal expression about the label of pesticide used was verified through observation of its container.

### 2.4. Study Variables and Scoring

Variables on the knowledge level of farmers were collected and scored 0 or 1. They were then aggregated into an overall “knowledge variable” and this aggregated variable on knowledge was classified as adequate (13–17 score) or inadequate (0–12 score) (Appendix A).

Likewise, the attitude and perception of farmers about pesticide policy and market management had 14 variables with total 24 scores where 1–3 scores were given to each variable based on the perceived relative weightage by the researcher team. Based on the median value, farmers were then considered as having favorable (17–24 score) or unfavorable (0–16 score) perception (Appendix B).

The practice was scored based on 17 variables, segregated into 3 domains: Practice during purchasing (four variables), practice during mixing (four variables), and spraying and practice during storage and disposal (nine variables). Each variable scored 1 if the practice conformed to safety requirements or 0 if it did not. Scores in each domain were aggregated, and taking the reference of its median value, categorized into safe (≥median) and unsafe (<median) practices. Possible confounders such as sex, age, caste/ethnicity, and education were collected.

Out of 792 sampled farmers, 790 participated in the study. The farmers exclusively using chemical pesticides (n = 663) were included in further analysis of practice. However, the knowledge and attitude/perception related questions were answered by 790 farmers.

### 2.5. Data Analysis

Data was entered into CSPro 7.3 software and analyzed using IBM SPSS 25. Descriptive statistics were generated and proportions were compared using Pearson’s Chi Square Test. Multinomial Logistic Regression was then used to assess the association among different variables and to calculate adjusted odds ratio. Statistical significance was determined at *p* < 0.05.

## 3. Results

### 3.1. Socio-Demographic Characteristics of Farmers

Out of the 790 farmers, the mean age of farmers was 46.04. More than half of farmers (53.7%) belonged to the age group of 30–50 years, 85 below 30, and 281 above 50. Female farmers comprise 51%. Most of them were from *Brahmin* and *Chettri* ethnicities, followed by indigenous communities *(Tharu, Magar, Tamang, Newar*, *and Chepang).* More than two-thirds of farmers (68.1%) were able to read and write, and most had attended some level of formal education (Table 1).

### 3.2. Use of Pesticide by the Farmers

Among the farmers participating in the study, 663 (84.0%) exclusively used chemical pesticide, while 28 farmers (3.5%) used botanical bio-pesticides only. The rest (12.5%) used both botanical and chemical pesticides in agriculture (Figure 1).

Among the 663 farmers who exclusively used chemical pesticides, 62% had no idea about the toxicity and label of pesticides on the pesticide container. Slightly more than one-fifth (20.7%) of farmers used yellow labeled pesticides, which are highly toxic with lethal dose 51–500 mg/kg. Two percent of farmers used banned pesticides indicated by red labels (extremely toxic with lethal dose 1–50 mg/kg). Blue and green labelled pesticides were used by 6.0 and 9.4 percent of farmers, respectively.

Among the 663 farmers using chemical pesticides, 60.8% had been using it for more than a decade. Further, most (96%) farmers took advice from a nearby agro-vet (pesticide retailers) on matters related to pest problems and the choice and use of pesticide.

### 3.3. Knowledge of Farmers about Safe Handling of Chemical Pesticides

More than 90 percent of farmers had knowledge about the importance to store pesticides away from the reach of children and animals and about safety clothes while spraying pesticide (Table 2). Knowledge to check the manufacture and expiry date of pesticides was found to be high (84.9%), while checking for the label and information about waiting period before harvest during purchase were low (30% and 32%, respectively). Similarly, only a small proportion of farmers knew the procedure of triple rinsing to clean the pesticide container after the spray (14.4%).

### 3.4. Attitude and Perception of Farmers about the Role of Government, Consumers, and Farmers to Reduce the Use of Chemical Pesticides

Eighty percent of the farmers believed that the government should discourage the irrational use of chemical pesticides by providing subsidies to farmers adopting organic farming and Integrated Pest Management (IPM) methods, where the use of pesticides is prohibited and minimal, respectively, and establishing separate market and fixing the prices for these produces (Table 3). The necessity of conduction of consumer awareness programs on detrimental effects of pesticide contaminated foods and prevention measures was shared by 77 percent farmers as the role of government. Regular supervision and monitoring by the government officials on the hazardous use of pesticides were opined by 63 percent farmers. Similarly, 51 percent farmers expressed their views on the role of government that they should strictly check the open border of the country to discourage the illegal import, sales, and use of unregistered and hazardous pesticides. Likewise establishment and effective functioning of pesticide residue measurement laboratory was cited by 41 percent farmers as one of the essential roles of government to minimize pesticides. Addressing the issue of pesticide through policy guidelines was pointed out by 77 and 37 percent farmers, respectively. Nearly three-fourth of the farmers perceived consumers should be more cautious towards their health and 18 percent said consumers should also inquire about pesticide use in foods they buy from the market. More than 85 percent farmers thought they also have the responsibility to promote organic products through their willingness and innovativeness to practice alternative approaches to chemical pesticides in agriculture.

### 3.5. Practice of Chemical Pesticide Use

Practice of chemical pesticide use by farmers was organized into 3 domains; practice during purchase, practice during mixing and spray, and practice of storage and disposal of chemical pesticides.

Three-fourths of the farmers reported that they checked manufacture and expiry date, while less than one-fourth observed the label of pesticide during purchase (Table 4). Fifty four percent of the farmers used protective equipment during spray. Most commonly used protective equipment were masks (79.7%), full sleeved clothes (62.1%), and gloves (44.4%). Most of the farmers stored the chemical pesticides away from the reach of children and animals in a separate place. Pesticides were packed tying with a plastic bag and placed at house ceilings to limit the access by children and animals. There was no lockable container to store pesticides among households. Less than thirty percent of farmers considered safety and environment during disposal of pesticides.

### 3.6. Scores on Knowledge, Attitude/Perception and Practice of Farmers about Chemical Pesticide Use and Its Safe Handling

The knowledge, attitude/perception, and the total practice scores were dichotomized into median and above or below the median as adequate/inadequate knowledge, favorable/unfavorable attitude/perception, and safe/unsafe practice, respectively. Accordingly, forty percent of farmers had adequate knowledge about the safe handling of pesticide, and a similar proportion also practiced safe handling. A similar proportion of farmers had favorable perception towards the role of local government, consumers, and themselves to reduce the use of pesticides (Figure 2).

### 3.7. Association of Safe Practice of Chemical Pesticides with Farmers’ Knowledge, Attitude/Perception and Socio-Demographic Factors

Out of the six variables studied, positive association with the practice of farmers on safe handling of pesticides was observed with knowledge about safe practice and perception of farmers about market management, gender, and education (Table 5). Farmers who had adequate knowledge were 8.3 times more likely to practice safe purchasing, four times more likely to practice safe mixing and spraying, and two times more likely to safely store and dispose. Similarly, perception of farmers about chemical pesticide policy and market management was significantly associated with the practice of farmers. Likewise, educated and male farmers were more likely to practice safer use of pesticides at different stages of purchase, mix and spray, and storage and disposal than uneducated and female farmers. There was no significant association between age of farmers and their caste/ethnicity with their practice of adoption of safety measures while handling pesticides, so these variables were not included in the final analysis presented in Table 5.

### 3.8. Acute Pesticide Poisoning Experienced by Farmers (n = 663)

Nearly, one-fifth of farmers (18.7%, n = 124) had experienced one or more acute symptoms of health problems after handling pesticides during the previous 12 months, which they related to the use of chemical pesticides. Among them, dizziness and headache (n = 74), skin allergies (n = 66), and burning of eyes (n = 35) were the most common symptoms. Others reported nausea/vomiting (n = 34), blurred vision, and swelling of body and muscle cramps (n = 20) (Figure 3). Farmers with unsafe practice of pesticide handling were two times more likely to suffer from acute poisoning (COR = 2.2, 95% CI = 1.4–3.3) (Table 6). Most (89.5%) of the farmers perceived these symptoms as normal or usual phenomena while handling pesticides, and therefore ignored health facility visits. There were no significant associations between acute health symptoms experienced by farmers and their age, sex, and education (Table 5).

## 4. Discussion

The study assessed different aspects of chemical pesticides use by farmers of the Chitwan district, and the self-reported health problems experienced by them. It addresses the research gap on factors contributing to safe and unsafe practice at different stages of pesticide handling namely during purchase, during mixing and spraying, and during storage and disposal. Additionally, factors such as perception of farmers towards the market management or the role of local government and consumers to minimize pesticide are less explored, and hence this research article could provide a scientific basis to advocate for enabling an environment for the reduction of irrational use of chemical pesticides in agriculture farming.

### 4.1. Rampant Use of Chemical Pesticide in Chitwan

An important revelation of the study is that 84 percent of the farmers in Chitwan are currently using exclusively chemical pesticides. Less than four percent are using botanical pesticides. It is likely that farmers are using botanical pesticides in small scale farming and in vegetable production for self-consumption. Chemical pesticides are widely used in commercial agriculture products, which are consumed by the larger consumers from the local and distant markets. As the study revealed, more than 60 percent of the farmers have been using chemical pesticides for more than 10 years, which means that farmers and general population have long term exposure to chemical pesticides. Two percent of the farmers use chemical pesticides labeled red which are banned in Nepal due to their extreme hazardous effect in health. This is a matter of serious concern that these pesticides are still available in the market and used by some farmers, as also suggested by other studies from Nepal [23]. This indicates towards an urgent need for monitoring the pesticide market.

### 4.2. Safe Handling of Pesticides by the Farmers

We studied the practice of pesticide use in three parts—during purchase; during mixing and spray; and storage and disposal. Additionally, in all the three possible stages of exposure, they did not practice safe handling of the chemical pesticides.

Label of pesticides is a critical marker of hazardousness and toxicity of pesticides. Only a quarter of farmers observed the label and toxicity of pesticide during purchase and 16.9% observed waiting time of the pesticide during purchase indicates the unawareness of farmers about safety provisions during purchase of pesticides. Similar findings have been observed in Kavrepalanchok [24] and Chitwan [25], where low levels of education and awareness among farmers posed difficulty to farmers to read the instructions in the international language. A study from Kuwait also depicted a similar scenario, where 70% of farmers did not go through the instructions in the pesticide container, and education level was associated with it [26].

During mixing and spraying of the pesticides, less than half of the farmers followed safe practices. Wearing protective clothes is one of the common safety measures. The study found that 54 percent of the farmers used any of the protective clothes during mixing and spraying of the pesticides, similar to that of Kuwait (58%) [26]. However, safe practice is better than in Northern Tanzania [27], where less than 10 percent of farmers were completely covered during spray.

Less use of protective clothes in Chitwan district might be due to the lack of awareness among the farmers, lack of availability when needed, discomfort due to hot and humid climate, and possibly might be due to cost factor. Similar findings have been shown by other studies [28,29,30], where cost, discomfort, and tropical factors were sought as major reasons for not using PPE. In Chitwan, Nepal, where the climatic condition is very hot, the cost of PPE ranges from NRs. 3500 to 5000 and are often not available in the local market. Government should consider programs to increase the availability and accessibility of farmers to personal protective equipment.

Practice of safe storage is followed by the majority of the farmers in Chitwan district, as the study revealed that 90% of them stored pesticides in a separate place away from access of children and animals. Storage practice in Chitwan is better than in Sri Lanka where 76% of farmers stored inside the house or immediately outside the house [31]. Nearly three-fourth of farmers dumped pesticide containers without consideration of their hazardous impact on the environment and humans, similar to that of Southwest Nigeria (72%) [32], probably due to lack of awareness and ignorance. Indeed, farmers have been reported to be inadequately informed about health and environmental hazards due to unsafe disposal of pesticide containers [33].

### 4.3. Factors Affecting Safe Practice of Pesticides

Overall safe practice of pesticides during purchase, spray, storage, and disposal was significantly associated with gender, literacy status, knowledge, and perception of the farmers in multivariate logistic regression. Male, literate farmers were more careful during purchase compared to female and illiterate farmers. Association of literacy status of farmers and knowledge with safe practice during purchase can be logically explained, as done by a systematic review conducted between 1999 and 2019 with 121 articles [34]. Additionally, from the present study, gender, and literacy status were found significantly associated, where 73.0% males were literate as compared to 63.3% literacy among females, therefore it is more likely that gender could be a confounding variable for higher knowledge among males with regard to pesticides. Besides, high exposures to the media and outside environment for males could also be a potential explanation, as found in a Chinese study [35].

In this study, knowledge of farmers about pesticide handling is strongly associated with safe pesticide use practice for all the three stages (purchase, use, and disposal). This finding of the study is consistent with the results of many other studies conducted in different countries [21,29,36], indicating the need for various programs to increase knowledge of farmers about safe practice of pesticides.

We explored the perception of farmers about the existing situation of pesticide use and their view on the role of different stakeholders, which is crucial in promoting rational and safe use of pesticide. We found that the overall perception of the farmers is positive and favorable to promote rational use of chemical pesticides. Farmers are concerned about the role of government, and have expressed that the government should provide subsidy and provide a separate market for organic/IPM products. It is encouraging that 86% of farmers are willing to search for alternatives to chemical pesticides and 88% of them prefer organic farming. These findings are similar to the studies [37,38,39] which have shown policies and legislation to support market returns and information acquisition had a significant positive influence on standardized pesticide application.

### 4.4. Health Effects on the Farmers

The study revealed that one-fifth of the farmers had experienced one or more acute health problems related to pesticide during the previous year. Among them, dizziness and headache, skin allergies, and burning of eyes were the most common symptoms. These are most common acute health problems due to exposure to chemical pesticides, reported elsewhere in Nepal [13,40] as well and other countries [41,42,43]. The acute problems were significantly higher among those with unsafe spray practice, which is similar to that of other parts of Nepal [13] indicating the need to promote safe handling of pesticides by the farmers. Furthermore, the majority of farmers with acute health symptoms did not attend any health facility accepting that such health problems are normal to the farm workers, a finding common to other developing countries as well [44].

### 4.5. Limitation and Strengths

Field verification on buying and spraying related practice was not feasible. However, the paper has firmly assessed the use of pesticides and its storage through cross questioning and observation as far as possible. Additionally, health problems experienced by the farmers were based on recall for one year period and can be affected by recall bias. We tried to reduce this bias through probing on the types, severity of symptoms, and how they responded to it. Besides, finding out the perceptions of farmers towards chemical pesticide policy and market management could provide a new outlook to motivate farmers towards safe practice along with the enhancement of their knowledge.

## 5. Conclusions

There was a high use of chemical pesticide in agriculture in Chitwan District, Nepal. It is concluded that knowledge about pesticide handling and favorable perception/attitude on pesticide policy and market management are the predictors of safe use of pesticide. Safe pesticide handling practices was found to be significantly associated with reduced acute pesticide poisoning in Nepal. There was no significant associations between age, sex, and education of farmers with their health symptoms.

## Figures and Tables

**Figure 1 ijerph-18-04194-f001:**
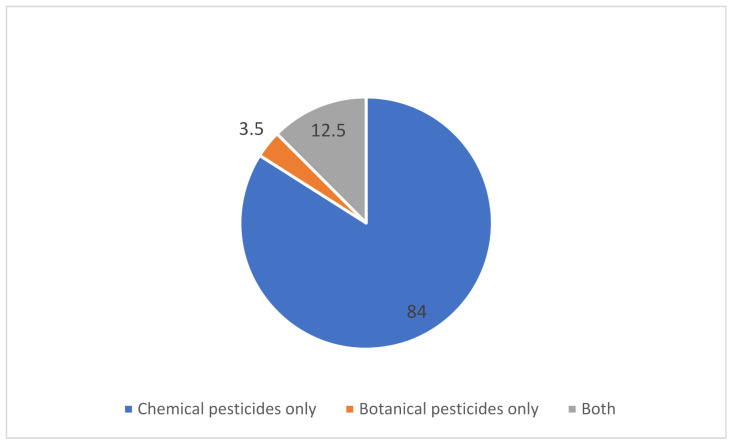
Types of pesticides used by farmers.

**Figure 2 ijerph-18-04194-f002:**
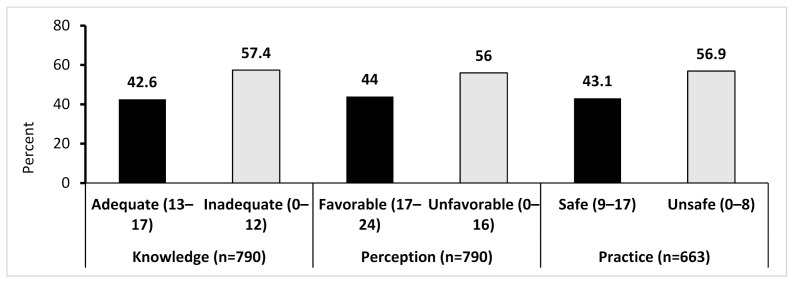
Knowledge, practice and perception of farmers about pesticide use and its safe handling.

**Figure 3 ijerph-18-04194-f003:**
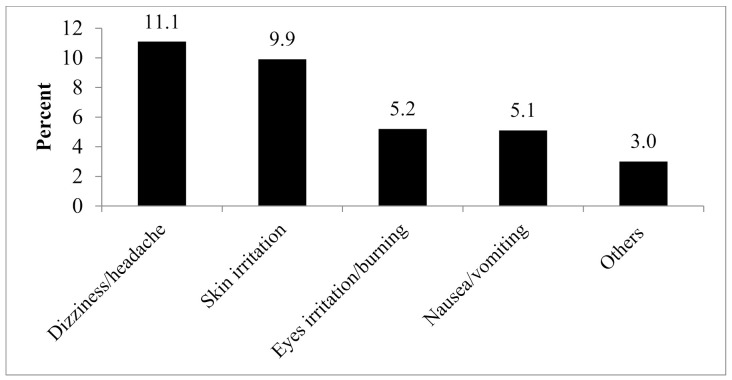
Acute pesticide poisoning experienced by farmers.

**Table 1 ijerph-18-04194-t001:** Socio-demographic characteristics of the farmers (n = 790).

Descriptions	Frequency	Percent
Gender		
Male	389	49.2
Female	401	50.8
Age group		
18–29 years	85	10.8
30–50 years	424	53.7
51 years and above	281	35.6
Caste/ethnicity		
Brahmin/Chhetri	500	63.3
Indigenous *	203	25.7
Dalits and others	87	11.0
Education		
Can read and write (literate)	538	68.1
Cannot read and write or only name	252	31.9

* Tharu, Magar, Tamang, Newar, Chepang.

**Table 2 ijerph-18-04194-t002:** Knowledge of chemical pesticide use of farmers in the domains of purchase, mixing and spraying, and storage and disposal (n = 790).

Descriptions	Number	Percent
During purchase		
Check manufacture and expiry date	670	84.9
Check whether the bottle is sealed	615	77.9
Observe the labels of pesticide	239	30.3
Check the indication about waiting period	252	31.9
During mixing and spray		
Mix pesticide considering the dose indicated	366	46.4
Mix pesticide away from water sources	578	73.3
Check the container if it is leaking	483	61.2
Wear protective clothes while spray	736	93.2
Spray considering the wind blowing direction	621	78.7
Spray at the right time of the day (evening and in the morning after the dew is dried out)	562	71.2
Maintain at least 1 m distance between nozzle to body	421	53.4
Spray at the right stage of the crop development	380	48.2
Take caution not to eat, drink, or smoke during spray	677	85.8
During storage and disposal		
Store in a dry place	542	68.6
Store pesticide in a separate place away from children and animals	744	94.3
Wash spray tank after use with triple rinsing method	114	14.4
Dispose container safely with the consideration of the environment (bury in an unused area)	396	50.1

**Table 3 ijerph-18-04194-t003:** Attitude and perception of farmers about chemical pesticide policy and market management (n = 790).

Description	Number	Percent
Role of the government		
Provide subsidy for promoting organic/IPM farmers	636	80.5
Establish separate market and fix a price for IPM/organic products	629	79.6
Conduct consumer awareness programs	610	77.2
Regular supervision and monitoring of pesticide use	499	63.2
Check open border for unregistered and hazardous pesticides	400	50.6
Establish pesticide residue measurement laboratory	324	41.0
Control import and promote local farmers products	413	52.3
Develop policy guidelines for market management	291	36.8
Role of consumers		
Show concern about pesticide use in vegetable market	138	17.5
Prefer organic product	397	50.3
Select vegetable based on season, color, and size	313	39.6
Be conscious about health effect of pesticides	559	70.8
Role of farmers		
Have willingness to practice organic farming	700	88.6
Search for alternative to chemical pesticides	686	86.8

**Table 4 ijerph-18-04194-t004:** Practice of chemical pesticide use by farmers (n = 663).

Descriptions	Number	Percent
During purchase		
Check the manufacture and expiry date	489	73.8
Check whether the bottle is sealed	436	65.8
Observe the labels of pesticide	153	23.1
Check the indication about waiting period	112	16.9
During mixing and spray		
Mix pesticide considering the dose indicated	229	34.5
Mix pesticide away from water sources	448	67.6
Check the container if it is leaking	281	42.4
Wear protective clothes during spray	359	54.1
Spray considering the wind blowing direction	374	56.4
Spray at the right time of the day (evening and in the morning after the dew is dried out)	241	36.3
Maintain at least 1 m far from nozzle to body	189	28.5
Spray at the right stage of the crop development (not during flowering stage)	161	24.3
Take caution not to eat, drink, or smoke during spray	309	46.6
During storage and disposal		
Store in a dry place	497	75.0
Store pesticide in a separate place (away from children and animals)	601	90.6
Wash the spray tank after use with triple rinsing method	346	52.2
Dispose the container safely with the consideration of the environment (bury in an unused area)	194	29.3

Note: The figures in the table indicate number and percentage of farmers who practiced the safety measures.

**Table 5 ijerph-18-04194-t005:** Association of farmers’ pesticide handling practice with their knowledge and attitude/perception (n = 663).

Descriptions	During Purchase			During Mixing and Spray			During Storage and Disposal		
	Safe Practicen (%)	COR (95% CI)	AOR (95% CI)	Safe Practicen (%)	COR (95% CI)	AOR (95% CI)	Safe Practicen (%)	COR (95% CI)	AOR (95% CI)
Knowledge of farmers about safe handling of pesticides									
Adequate	141 (52.4)	11.2 (7.4–17.2) *	8.3 (5.0–13.8) *	176 (65.4)	6.6 (4.7–9.4) *	3.9 (2.5–5.9) *	177 (65.8)	3.5 (2.5–4.9) *	2.4 (1.6–3.6) *
Inadequate	35 (8.9)	1	1	87 (22.1)	1	1	138 (35.0)	1	1
Perception of farmers about chemical pesticide policy and market management									
Favorable	123 (44.7)	5.1 (3.5–7.4) *	1.7 (1.1–2.8) *	175 (63.6)	5.9 (4.2–8.3) *	3.0 (2.0–4.5) *	173 (62.9)	2.9 (2.1–4.0) *	1.7 (1.1–2.5) *
Unfavorable	53 (13.7)	1	1	88 (22.7)	1	1	142 (36.6)	1	1
Gender									
Male	114 (34.7)	2.3 (1.6–3.3) *	2.0 (1.3–3.1) *	176 (53.5)	3.2 (2.3–4.5) *	3.3 (2.2–4.8) *	172 (52.3)	1.4 (1.0–1.9) *	1.2 (0.8–1.7) *
Female	62 (18.6)	1	1	87 (26.0)	1	1	143 (42.8)	1	1
Education									
Can read and write (literate)	160 (36.2)	7.2 (4.2–12.5) *	6.8 (3.8–12.3) *	203 (45.9)	2.2 (1.6–3.2) *	1.7 (1.1–2.6) *	241 (54.5)	2.3 (1.7–3.3) *	2.0 (1.4–2.9) *
Cannot read and write/only name	16 (7.2)	1	1	60 (27.1)	1	1	74 (33.5)	1	1

* *p* < 0.05; COR: Crude odds ratio; AOR: Adjusted odds ratio.

**Table 6 ijerph-18-04194-t006:** Association of acute health symptoms with safe handling practice (n = 663).

Descriptions	Acute Health Symptoms	
	Yes n (%)	COR (95% CI)
Practice of farmers about safe handling of pesticides		
Unsafe use of pesticide	89 (23.6)	2.2 (1.4–3.3) *
Safe use of pesticide	35 (12.2)	1
Age of farmers		
Less than 40 years	38 (17.8)	1.1 (0.7–1.7)
40 years and above	86 (19.1)	1
Sex of farmers		
Male	61 (18.5)	1.0 (0.7–1.5)
Female	63 (18.9)	1
Education of farmers		
Can read and write (literate)	75 (17.0)	1.3 (0.9–2.0)
Cannot read and write/only name	49 (22.2)	1

* *p* < 0.05; COR: Crude odds ratio.

## Data Availability

The data are not publicly available due to privacy.

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
