# Peer review of "Factors Associated with Practice of Chemical Pesticide Use and Acute Poisoning Experienced by Farmers in Chitwan District, Nepal"

_ijerph, 2021, doi:10.3390/ijerph18084194_

Round 1

Reviewer 1 Report

The authors present a detailed study on the practises embraced by farmers in a district of Nepal, concerning the pesticides' use. It is a manuscript with merit for publication. I have limited comments, which if considered the manuscript will be further improved prior to acceptance.

  1. Page 1, line 39, remove "of consumption"
  2. Page 4, line 167 after percent add "of", lines 168-169, i would suggest the use of personal protection equipment (initially) and then authors can specify the part (clothes, inner-outer etc.). 
  3. A general remark in the tables is that the terms number and description need further elaboration. For example, in Table 2, the description might include opinion, expressed by people interviewed. Hence, such info should be added. 
  4. Page 6, line 197, one-fourth.
  5. Page 10, line 265, please consider the general ranking of pesticide use period-duration and justify the "very long time". A suggestion is to use the phrase "long-term".
  6.  Page 10, line 291, the availability of protective equipment should be elucidated more.  Same page line 299, this point needs additional data. To place away, from children, pesticides is not always safe. Were the containers or the dedicated facilities locked with restricted access? Such data are indispensable. 
  7. Page 11, line 305, please clarify the phrase "ill-informed"
  8. An additional comment to regard in future studies, is the discrimination of categories of pesticide users (e.g., operators, mixer-loaders etc.) if it can be evaluated, of course. 

Author Response

Point 1: Page 1, line 39, remove "of consumption"

Response to point 1: Removed.

Point 2: Page 4, line 167 after percent add "of", lines 168-169, i would suggest the use of personal protection equipment (initially) and then authors can specify the part (clothes, inner-outer etc.). 

Response to point 2: Addressed in 239 and 241 line, page 8.

Point 3: A general remark in the tables is that the terms number and description need further elaboration. For example, in Table 2, the description might include opinion, expressed by people interviewed. Hence, such info should be added. 

Response to point 3: Addressed from line 212-224, page 7.

Point 4: Page 6, line 197, one-fourth.

Response to point 4: Edited, page 8, line 237

Point 5: Page 10, line 265, please consider the general ranking of pesticide use period-duration and justify the "very long time". A suggestion is to use the phrase "long-term".

Response to point 5: The phrase long term has been used, page 12, point 4.1.

Point 6: Page 10, line 291, the availability of protective equipment should be elucidated more.  Same page line 299, this point needs additional data. To place away, from children, pesticides is not always safe. Were the containers or the dedicated facilities locked with restricted access? Such data are indispensable. 

Response to point 6: The details of protective equipment have been added in 239 and 241 line, page 8. The storage place was separate, for example, pesticides were packed tying with a plastic bag and placed in the house ceilings to limit the access by children and animals, but not in a lockable container. Addressed in line 241-245, page 8.

Point 7: Page 11, line 305, please clarify the phrase "ill-informed"

Response to point 7: It has been replaced with inadequately informed, page 13, point 4.2.

Point 8: An additional comment to regard in future studies, is the discrimination of categories of pesticide users (e.g., operators, mixer-loaders etc.) if it can be evaluated, of course. 

Response to point 8: This is a good idea indeed. But the only mode of application in the study area was by individual farmers using spray tank manually. It is usually an individual household member who takes the responsibility to buy pesticides, mix and spray it using knapsack.

Reviewer 2 Report

Comments:

In this study, the authors investigated the factors associated with the practice of chemical pesticide use and potential adverse health outcome in a specific area. Overall, it brought concerning public health issues in the selected area, where 84% of the farmers were reported to exclusively use chemical pesticides, and even some of them were still using banned pesticides. This report has its own impacts on multiple fields including to the farmers, they should grow awareness towards the hazardous effects of pesticides, and to local policy makers, they should pay special attention to pesticide regulation, as well as the educations providing for farmers with safe application of pesticides.

Despite the importance of the work itself, this manuscript was not well written, the data analysis results were not well presented. For example, result 3.1 could be better depicted in a distribution bar graph, where the readers can easily see the distribution rather than being drowned in a bunch of numbers.  

Results 3.2 and 3.6, pie charts would be much more helpful to show the percentage differences.

-In the conclusion, which sex or which age group of farmers were more susceptible to health problem caused by pesticides?

Specific comments:

-Line 17, it should be X2.

-Line 34-36, this sentence could be more concise, e.g. deleting the exact number but the percentage, or vice versa.

-Line 39, delete “per hectare”.

-Line 46, please define FAO/WHO.

-Line 49, replace “But” with “However,”.

-Line 52, public health.

-Formula in 2.2 section is not clearly described, the mix of upper and lowercase letters are not explained in detail, what is Q and e? and the calculation results are incorrect.

-Line 139, OR=odd ratio

-Line 146, maybe a typo, it should be “two-thirds”.

-Line 181, please define “IPM”.

Author Response

Response to reviewer 2 comments

Point 1: Despite the importance of the work itself, this manuscript was not well written, the data analysis results were not well presented. For example, result 3.1 could be better depicted in a distribution bar graph, where the readers can easily see the distribution rather than being drowned in a bunch of numbers.  Results 3.2 and 3.6, pie charts would be much more helpful to show the percentage differences.

Response to point 1:

  1. Regarding the 3.1, a table has been added for clear presentation of number and percentage as per different categories (page 4).
  2. For results 3.2, a pie chart has been added on the types of pesticides (page 5). However for other different variables under 3.2, authors have discussed and decided that the presentation in text format would be fine because to present all the information in pie charts would require many pie charts in 3.2. One pie chart can reflect only one category of information.
  3. The same is true for 3.6 as well, however, it is reflected in bar graph to avoid the monotony of text.

Point 2: In the conclusion, which sex or which age group of farmers were more susceptible to health problem caused by pesticides?

Response to point 2: This information has been included in the conclusion, page 14. Also see table 6 where we have included other variables too. But the associations are shown insignificant. Therefore, shall we remove the sex, age and education variables from the table itself and write only in descriptions? After that table 6 will only contain association between safe handling of pesticide and acute health symptoms with COR.

Table 6. Association of acute health symptoms with safe handling practice (n=663).

Descriptions

Acute health symptoms

Yes

n (%)

COR

(95% CI)

AOR

(95% CI)

Practice of farmers about safe handling of pesticides

Unsafe use of pesticide

89 (23.6)

2.2 (1.4-3.3)*

2.1 (1.3-3.3)*

Safe use of pesticide

35 (12.2)

1

1

Age of farmers

Less than 40 years

38 (17.8)

1.1 (0.7-1.7)

40 years and above

86 (19.1)

1

Sex of farmers

Male

61 (18.5)

1.0 (0.7-1.5)

Female

63 (18.9)

1

Education of farmers

Can read and write (literate)

75 (17.0)

1.3 (0.9-2.0)

Can't read and write/ only name

49 (22.2)

1

Reviewer 3 Report

“Factors associated with practice of chemical pesticide use 2 and acute poisoning experienced by farmers in Chitwan 3 District, Nepal” written by Kafle and colleagues.

I found the subject and the results very interesting but the manuscript suffers seriously from lack of some information. The article is a descriptive (and not completely analytic) survey without giving more information about the pesticides used by the farmers as the main element of this investigation.

Abstract:

The abstract is somewhat general and does not reflect completely the findings of this study. Please revise it by shortening the less important items and concentrating more on the main findings and conclusion of present study.

Main text:

There is a lack of information about the type of pesticides used by the farmers, toxicity, mode of application, pesticide residues, residual activity of pesticides, etc. The authors state about the use of botanic and chemical insecticides by the farmers. All of these pesticides are toxic for human. Even among the chemical pesticides, the level of toxicity is not equal among different groups of pesticides. Therefore, the toxicity and side-effect of these chemicals are not similar.

In my opinion, two main findings of this study rely mainly on the “Socio-demographic characteristics of farmers” (including knowledge, attitude and perception) and “mode of application of the pesticides” and to investigate if there is a significant correlation between these factors. For this purpose, I suggest arranging the main text according these two main sub-sections and develop more by giving the missing information as follow:

  1. The information about the insecticides used by the farmers. The authors tried to evaluate the hazardous effects associated with pesticide use among the farmers while the type of insecticides used by the farmers, mode of application, toxicity and etc. are not similar. As mentioned, the toxicity of pesticides depends on several factors such as the type of pesticide, mode of action (direct contact, spraying, fumigation, etc.), volume used, pesticide residues and etc.

  1. Have farmers been previously trained about pesticide use, their toxicity and probable side effects?

  1. How the farmers select the pesticides? Is it based on a governmental protocol or it is a self-choice?

  1. The authors explained briefly about the socio-demographic characteristics such as the age, level of education, etc. but I think that they can develop this sub-section by giving more information about the component of these elements. For instance, the farmers with what age group, education level or gender pay more attention for accurate pesticide application?

  1. Based on which characters, the authors could evaluate the attitude and perception of the studied farmers?

Author Response

 “Factors associated with practice of chemical pesticide use 2 and acute poisoning experienced by farmers in Chitwan 3 District, Nepal” written by Kafle and colleagues.

Point 1. I found the subject and the results very interesting but the manuscript suffers seriously from lack of some information. The article is a descriptive (and not completely analytic) survey without giving more information about the pesticides used by the farmers as the main element of this investigation.

Response to point 1: It has been addressed in page number 5, 3.2.

Point 2. The abstract is somewhat general and does not reflect completely the findings of this study. Please revise it by shortening the less important items and concentrating more on the main findings and conclusion of present study.

Response to point 2: The phrasing of words and texts in abstract have been reworked and the feedback has been implemented.

Point 3: Main text: There is a lack of information about the type of pesticides used by the farmers, toxicity, mode of application, pesticide residues, residual activity of pesticides, etc.

Response to point 3:

  1. The information on types of pesticides was obtained only with respect to chemical or botanical.
  2. The information of toxicity has been addressed in page number 5, 3.2, line 185-191.
  3. The only mode of application in the study area was by individual farmers using spray tank manually.
  4. No data was obtained with respect to pesticide residues and residual activity of pesticides.

Point 4: The authors state about the use of botanic and chemical insecticides by the farmers. All of these pesticides are toxic for human. Even among the chemical pesticides, the level of toxicity is not equal among different groups of pesticides. Therefore, the toxicity and side-effect of these chemicals are not similar.

Response to point 4: This is a good idea. But in this study, we do not have sufficient data to analyse the health symptoms as per the exposure to different levels of toxicity of pesticides.

Point 5: In my opinion, two main findings of this study rely mainly on the “Socio-demographic characteristics of farmers” (including knowledge, attitude and perception) and “mode of application of the pesticides” and to investigate if there is a significant correlation between these factors. For this purpose, I suggest arranging the main text according these two main sub-sections and develop more by giving the missing information as follow:

Response to point 5: The only mode of application in the study area was by individual farmers using spray tank manually. Therefore, no analysis could be done based on difference in mode of applications. But different aspects of mode of application such as if caution was taken towards the direction of wind, spraying time in a day, if pesticide is sprayed during flowering stage, spraying with the use of protective clothes or not were dealt as specific indicators under 'Practice variable'. Please see page 6, table 2.

Point 6: The information about the insecticides used by the farmers. The authors tried to evaluate the hazardous effects associated with pesticide use among the farmers while the type of insecticides used by the farmers, mode of application, toxicity and etc. are not similar. As mentioned, the toxicity of pesticides depends on several factors such as the type of pesticide, mode of action (direct contact, spraying, fumigation, etc.), volume used, pesticide residues and etc.

Response to point 6:

  1. The information on types of pesticides was obtained only with respect to chemical or botanical or both.
  2. The information of toxicity has been addressed in page number 5, 3.2, line 185-191.
  3. The only mode of application in the study area was by individual farmers on a manual basis using knapsack.
  4. No data was obtained with respect to pesticide residues and residual activity of pesticides.

Point 7: Have farmers been previously trained about pesticide use, their toxicity and probable side effects?

Response to point 7: The component of training and its impact on knowledge and practice will be assessed comparing with baseline and mentioned in another manuscript. This manuscript will not cover training aspect.

Point 8: How the farmers select the pesticides? Is it based on a governmental protocol or it is a self-choice?

Response to point 8: More than 90% farmers reported buying pesticides based on the suggestions by pesticide retailers (this has been mentioned in results 3.2, line 193-195).

Point 9: The authors explained briefly about the socio-demographic characteristics such as the age, level of education, etc. but I think that they can develop this sub-section by giving more information about the component of these elements. For instance, the farmers with what age group, education level or gender pay more attention for accurate pesticide application?

Response to point 9: This has been addressed in page 9 from line 271-273.

Point 10: Based on which characters, the authors could evaluate the attitude and perception of the studied farmers?

Response to point 10: Characters are mentioned in Appendix B, page 15

Round 2

Reviewer 2 Report

Please define the "IPM". It can be accepted with present form.

Author Response

Response to reviewer 2 comments_2nd round

Point 1: Line 17, it should be X2.

Response: Edited.

Point 2: Line 34-36, this sentence could be more concise, e.g. deleting the exact number but the percentage, or vice versa.

Response: Removed the percentage in the bracket.

Point 3: Line 39, delete “per hectare”.

Response: Deleted

Point 4: Line 46, please define FAO/WHO.

Response: The abbreviation form of FAO and WHO has been kept.

Point 5: Line 49, replace “But” with “However,”.

Response: Replaced

Point 6: Line 52, public health.

Response: replaced jeopardizing the health of public with 'public health'.

Point 7: Formula in 2.2 section is not clearly described, the mix of upper and lowercase letters are not explained in detail, what is Q and e? and the calculation results are incorrect.

Response: The meaning of Q and e is explained and calculated was done. The new farmers' sample became 792. We wrote about this in the methodology that only 790 participated in the study, line 153.

Point 8: Line 139, OR=odd ratio

Response: The full form was kept for OR.

Point 9: Line 146, maybe a typo, it should be “two-thirds”.

Response: Edited.

Point 10: Line 181, please define “IPM”.

Response: The abbreviation was used along with the addition of a few lines to reflect its meaning.